# Positive Orientation and Fatigue Experienced by Polish Nursing Students during the COVID-19 Pandemic: The Mediatory Role of Emotional Control

**DOI:** 10.3390/jcm11112971

**Published:** 2022-05-25

**Authors:** Ewa Kupcewicz, Kamila Rachubińska, Aleksandra Gaworska-Krzemińska, Anna Andruszkiewicz, Ewa Kawalec-Kajstura, Dorota Kozieł, Małgorzata A. Basińska, Elżbieta Grochans

**Affiliations:** 1Department of Nursing, Collegium Medicum, University of Warmia and Mazury in Olsztyn, 14 Zolnierska Street, 10-719 Olsztyn, Poland; 2Department of Nursing, Pomeranian Medical University in Szczecin, 48 Zolnierska Street, 71-210 Szczecin, Poland; kamila.rachubinska@pum.edu.pl (K.R.); grochans@pum.edu.pl (E.G.); 3Institute of Nursing and Midwifery, Medical University of Gdansk, M. Sklodowskiej-Curie Street 3a, 80-227 Gdansk, Poland; aleksandra.gaworska-krzeminska@gumed.edu.pl; 4Department of Basic Clinical Skills and Postgraduate Education for Nurses and Midwifes, Nicolaus Copernicus University in Torun, 1 Łukasiewicza Street, 85-821 Bydgoszcz, Poland; anna.andruszkiewicz@cm.umk.pl; 5Department of Internal Medicine and Community Nursing, Institute of Nursing and Midwifery, Faculty of Health Sciences, Medical College, Jagiellonian University, 30-688 Krakow, Poland; e.kawalec@uj.edu.pl; 6Medical College, J. Kochanowski University in Kielce, 25-369 Kielce, Poland; dkoziel@ujk.edu.pl; 7Department of Clinical Psychology, Faculty of Psychology, Kazimierz Wielki University, 85-064 Bydgoszcz, Poland; malbasinska@wp.pl

**Keywords:** COVID-19 pandemic, psychological problems

## Abstract

This study aimed to investigate the mediatory role of emotional control with respect to the control of anger, depression, and anxiety in the relationship between positive orientation and tiredness/fatigue occurring in a group of Polish nursing students during the COVID-19 pandemic. The study included 894 nursing students from six universities in Poland. A diagnostic survey was applied as the research method, and the data were collected using the Fatigue Severity Scale (FSS), the Courtauld Emotional Control Scale (CECS) and the Positive Orientation Scale (SOP). The mean participant age was 20.73 years (SD = 1.81). More than half of the students in the study showed a low level of positive orientation. Correlational analyses revealed a significant negative correlation between positive orientation and tiredness/fatigue experienced by the students participating in the study (r = −0.336; *p* < 0.001), and correlation between positive orientation and the overall emotional control index (r = −0.317; *p* < 0.001), and the indices of control of anger (r = −0.154; *p* < 0.01), depression (r = −0.376; *p* < 0.001), and anxiety (r = −0.236; *p* < 0.01). Analysis of the results also revealed the occurrence of significant, positive links between the controlled emotions and their components and the tiredness/fatigue experienced by nursing students. It is important to take action associated with the prevention of tiredness/fatigue among students and to reinforce a positive orientation and the capacity to control emotions to effectively minimize the effects of the COVID-19 pandemic on nursing students.

## 1. Introduction

Several studies have been conducted on student adaptation [1], academic stress [2], depression [3], motivation [4] and positive psychology constructs, i.e., satisfaction with life [5], hope [6] and self-esteem [7]. However, few scholars have pursued studies aimed at examining aspects of human functioning which are associated with positive orientation and its adaptive importance in connection with the COVID-19 pandemic [5,7,8,9,10]. 

### 1.1. Positive Orientation

Caprara suggested that positive orientation is the basic tendency to notice positive aspects of life, experience and oneself and to regard them as important. It is a latent variable of a higher order, combining three components: self-esteem, optimism and life satisfaction [5,7,11]. Investigation of positive orientation was inspired by an attempt to determine the variables which are the inverse of the depressive cognitive triad proposed by Beck, who claimed that, unlike healthy people, individuals with depression had a negative opinion about themselves, the world and the future [12]. It has been confirmed in Canadian, German and Japanese studies that a positive orientation can be regarded as a “good functioning syndrome”, which is positively correlated to an assessment of health status. A correlation between positive orientation and self-efficacy was confirmed. These convictions have an impact on the cognitive and emotional component of subjective well-being, i.e., positive thinking and feelings of happiness. They also correspond to the difference between positive and negative affect [13,14,15,16]. A positive orientation encourages people to interpret their lives in a positive manner. Student mental health, social adaptation and positive orientation have become important issues in academic circles recently [17]. 

### 1.2. Emotional Control

Studies conducted by Bidzan et al. [18] demonstrated the positive value of the possibility of expressing one’s emotions in the face of a disaster. An individual’s personal resources play a key role in the expression of emotions. These resources include self-control, self-efficacy, resourcefulness, sense of humor, valuation and coping with events of concern [19,20,21]. Future medical professionals are expected to show mental resilience, with the experience of emotions and their specific expression as significant elements. According to Averill’s theory, the diversity of emotions felt by an individual is a consequence of the individual having mental and emotional responses to events [22]. Experiencing emotions is usually associated with somatic changes, mimic and pantomimic expressions and specific behavior [23]. Emotions are associated with self-control, which is defined as behavior compliant with the norms adopted by an individual or social norms [24]. Absorbing the patterns of emotions dominating in a social environment is sometimes compared with learning to perform roles [25,26,27]. Literature reports stress that suppression of emotions is an adverse phenomenon, as it can lead to intensification of emotions or to their persistence as emotional tension [20]. Taking preventive action can involve establishing a set of factors harmful to health, including the style of emotional expression. The COVID-19 pandemic has had an adverse impact on people’s lifestyles, with serious consequences for their mental health, especially among young people [28,29,30]. Challenges faced by institutions because of the COVID-19 pandemic have accelerated the transformation in teaching methods in all education sectors [31]. Documenting and analyzing the current outcomes of the changes is important for learning and implementing new pedagogic rules, leading to educational transformation, which can change the way we teach and learn forever [32,33]. 

### 1.3. Tiredness/Fatigue

Fatigue associated with lockdowns and online teaching is caused by overwhelming changes in the functioning of an individual, i.e., social isolation, lack of a sense of security, direct threat to health and unpredictability of what will happen. Fatigue can be a result of a combination of many factors of physical, mental and/or emotional origin [34]. Fatigue in the literature is understood to denote an unpleasant physical feeling with associated cognitive and behavioral symptoms and a lack of energy [35]. The longer and the more intense it is, the more disruptive it is with respect to everyday activities. Fatigue can be associated with multiple somatic and mental diseases, but it is also considered in relation to professional burnout or chronic fatigue. Professional burnout syndrome used to refer to professional work [36], but, over time, it has also become noticeable in students [37]. In students, it usually manifests as exhaustion, a cynical attitude to studying and learning, as well as a sense of incompetence and ineffectiveness [38,39]. Kutyło showed that burnout in students increases the probability of not attending classes, and also affects their marks and results in the sense of alienation and failure [40,41,42]. A change in the form of teaching during the COVID-19 pandemic has posed a threat to the teaching and learning process and to students’ mental health because of the extent of exhaustion or fatigue [33,43]. According to the findings of a study by de Oliveira Kubrusly Sobral et al., excessive use of video conferences may result in fatigue as it requires focusing more intensely than when attending a stationary event [33]. The use of a video chat makes its participants work harder on processing non-verbal signals, such as facial expression, tone of voice and body language [44], and they are more susceptible to fatigue with Zoom [45]. Online education forces a person to spend more time in front of the screen, reduces social contacts, creates cognitive and emotional issues and induces a sense of chronic fatigue [33,43,44,45].

The study aimed to investigate the mediatory role of emotional control, with respect to the control of anger, depression and anxiety, in the relationship between positive orientation and tiredness/fatigue occurring in a group of Polish nursing students during the COVID-19 pandemic. 

In view of the stated aim, the following research questions were formulated: Does positive orientation serve the role of a predictor of tiredness/fatigue occurring in the nursing students under study?Does emotional control, understood with respect to the control of anger, depression and anxiety, play a role in mediating in the relationship between positive orientation and tiredness/fatigue in nursing students?

## 2. Materials and Methods

### 2.1. Settings and Design

Nursing students were invited to participate in a survey that was conducted between 20 March and 15 December 2021 at the Pomeranian Medical University in Szczecin (*n* = 215; 24.05%), the University of Warmia and Mazury in Olsztyn (*n* = 175; 19.57%), the Medical University of Gdańsk (*n* = 143; 16.00%), the Nicolaus Copernicus University in Toruń, Collegium Medicum in Bydgoszcz (*n* = 171; 19.24%), the Jagiellonian University in Kraków (*n* = 132; 14.77%) and the Jan Kochanowski University of Kielce (*n* = 57; 6.38%). Nursing students who were up to 30 years of age, and gave informed consent to participate in the study, were enrolled. Those who failed to give such consent were excluded from the study. Having obtained the deans’ consent, researchers conducted the survey in direct contact with the students while maintaining a sanitary regime. Participation in the study was voluntary and anonymous. The respondents were informed that they could withdraw from the study at any time. The survey was conducted in groups of several students. The subjects were informed about the study objective and had an opportunity to ask questions and receive answers. It took approximately 15 min to complete the questionnaire. A total of 975 questionnaire sets were distributed, and 894 (91.69%) were returned. This study is part of a larger research project, which was approved (No. 3/2021) by the Senate Scientific Research Ethics Committee at the Olsztyn University in Olsztyn. It was conducted in accordance with the rules of the Helsinki Declaration. 

### 2.2. Research Instruments

A diagnostic survey method was applied and three standardized research tools were used to collect empirical data:The Fatigue Severity Scale developed by L.B. Krupp et al. [46,47];Emotion Control Scale—CECS developed by M. Watson, S. Greer [48];Positive Orientation Scale developed by Caprara et al. [16].

The participants also filled in a questionnaire with questions about the basic sociodemographic data associated with selected lifestyle elements. 

#### 2.2.1. The Fatigue Severity Scale (FSS)

The FSS (developed by L.B. Krupp et al.) is used to evaluate the impact of fatigue experienced by an individual. It comprises nine statements with a score of 1 to 7 on a Likert scale, which provide the basis for the determination of symptom severity. It covers both the physical and mental aspects of fatigue. One point means that the participant definitely disagrees with the statement, whereas 7 points indicates that he/she definitely agrees with it. The maximum score is 63. The higher the score, the higher the fatigue severity. The limit point denoting the normal level is not used in the FSS scale analysis, but it is assumed that a score of 36 or more points is indicative of significant clinical fatigue. The scale reliability is high: the internal consistency index (Cronbach *α*) is 0.88 for healthy individuals and the test-retest reliability is 0.84 [46,47]. 

#### 2.2.2. The Courtauld Emotional Control Scale (CECS)

The CECS scale (developed by M. Watson and S. Greer, Polish version by Z. Juczyński) is used to measure the subjective control of anger, anxiety and depression in difficult situations. It consists of three subscales, each of which includes seven statements on ways of showing anger, anxiety and depression. A participant specifies how often a specific way of showing their emotions occurs, on a four-point scale, from “hardly ever” (1 point) to “nearly always” (4 points). The score for each of the scales ranges from 7 to 28 points. The total emotion control index ranges from 21 to 84 points and is calculated by adding up the scores for the three subscales. It denotes an individual’s subjective conviction about their ability to control their responses to specific negative emotions. The higher the score, the higher the degree of emotional suppression. The scale reliability is high: the internal consistency index (Cronbach *α*) is 0.87 for the general emotion index, 0.80 for anger, 0.77 for depression and 0.78 for anxiety control [48].

#### 2.2.3. The Positivity Scale (SOP)

The positivity scale (developed by Caprara et al., Polish version by M. Laguna et al.) is used to measure the basic tendency to notice and appreciate positive aspects of life, experiences and oneself. The scale is made up of eight diagnostic statements. A participant states to what extent he/she agrees with each of them. The answers are given on a 5-point scale, from 1 to 5 (1—I definitely disagree, 5—I definitely agree). The raw scores lie within an interval from 8 to 40 points. The result is the total score; the higher it is, the higher the positive orientation level. A raw score of positive orientation is converted to standardized units on the sten scale and is interpreted in accordance with its properties. Scores between 1 and 4 sten are regarded as low, 5 and 6 as average, and those from 7 to 10 sten as above average. The P-scale in the Polish version has good psychometric properties. Its internal consistency (Cronbach *α*) ranges from 0.77 to 0.84 [16].

### 2.3. Statistical Analysis

A statistical analysis of the data was performed with the Polish version of STATISTICA 13 (TIBCO, Palo Alto, CA, USA). The mean (M), median (Me), standard deviation (SD), minimum and maximum (Min., Max.) and the confidence interval of the mean (95 CI) was calculated for all the variables. The immeasurable parameters were presented by means of the sample size and percentage. The correlations between the analyzed variables were determined by means of Pearson correlation coefficients (r). The correlation power interpretation was based on the Guilford classification, with: |r| = 0—no correlation; 0.0 < |r| ≤ 0.1—slight correlation; 0.1 < |r| ≤ 0.3—poor correlation; 0.3 < |r| ≤ 0.5—average correlation; 0.5 < |r| ≤ 0.7—high correlation; 0.7 < |r| ≤ 0.9—very high correlation; 0.9 < |r| < 1.0—nearly full correlation; |r| = 1—full correlation [49]. The mediatory effects were measured by the methodology developed by R.M. Baron and D.A. Kenny [50]. The Sobel test was used to verify the statistical significance of the mediation analysis model [51]. A *p* value < 0.05 was taken as the level of significance. 

## 3. Results

### 3.1. Participants

Altogether 894 nursing students participated in the study, including 822 women (91.5%) and 72 men (8.05%). There were 397 (44.41%) first-year students, 289 (32.33%) second-year students and 208 (23.27%) third-year students. The mean age of the participants was 20.73 years (SD = 1.81). The majority of the respondents (*n* = 621; 69.46%) lived with their family or with someone close. On average, they spent more than six hours working on a computer. A considerable reduction in their social contacts during the COVID-19 pandemic was mentioned by 40.27% (*n* = 360) of the respondents. Nearly all the respondents saw their health as good (*n* = 613; 68.57%) or very good (*n* = 257; 28.75%). Only 23.60% of the respondents stated that they had not reduced their physical activity because of the COVID-19 pandemic, whereas the others reduced it to various degrees (18.01%—slightly, 31.10%—to an average extent, 27.29%—considerably). The respondents chose various forms of activity: they usually mentioned walking (*n* = 184), general exercise (*n* = 80) and jogging (*n* = 71). They had 3.48 meals a day, on average (SD = 0.87). 

### 3.2. Variable Analysis 

A statistical analysis of the collected data was performed, and the mean values of the variables were calculated for the whole group of Polish nursing students. The mean positive orientation score was 26.87 (SD = 5.90) on a scale from 8 to 40 points, whereas the mean fatigue score was 37.93 (SD = 11.38) on a scale from 9 to 63 points, indicative of clinically significant fatigue. The analysis results revealed an overall emotional control score of 51.45 points (SD = 11.74), an anger control score of 16.66 (SD = 4.79), a depression control score of 17.87 (SD = 4.62) and an anxiety control score of 16.92 (SD = 4.85). The other results are shown in Table 1. 

Subsequent analyses involved the transformation of the raw score for positive orientation into standardized units on the sten scale. Individuals with low scores, who negatively perceived their abilities to cope with difficulties and adversities, accounted for the majority of the study population (51.23%). Scores regarded as average were noted for 29.08% of the participants, whereas under one fifth of them had high scores, which means that they perceived themselves as coping well and having a positive attitude to life (Figure 1). 

### 3.3. Mediation Analysis

For the purposes of this empirical study, a dependent variable, an independent variable and a mediating variable were identified. Tiredness/fatigue occurring in nursing students was adopted as the dependent variable, while positive orientation was adopted as an independent variable. The analyzed relationship between the variables was considered in the context of a mediatory effect. The mediatory variable (mediator) of the assumed relationship was the subjective emotional control understood in terms of control of anger, depression and anxiety. The first stage of the study analyzed the relationships between the variables. As the distributions of the variables were close to a normal distribution, parametric methods were applied. Using the r-Pearson coefficient, whether the variables of interest, at this point of the analyses, were significantly linked to each other with a linear relationship was assessed. Correlational analyses revealed a significant negative correlation between positive orientation and tiredness/fatigue experienced by the students participating in the study (r = −0.336; *p* < 0.001), and a correlation between positive orientation and the overall emotional control index (r = −0.317; *p* < 0.001) and the indices of control of anger (r = −0.154; *p* < 0.01), depression (r = −0.376; *p* < 0.001), and anxiety (r = −0.236; *p* < 0.01). Analysis of the results also revealed the occurrence of significant, positive links between the controlled emotions and their components and tiredness/fatigue occurring in nursing students. The presented correlation analysis results confirm the validity of the mediation model testing (Table 2). 

For the measurement of the mediatory effects, the methodology of three subsequent steps according to R.M. Baron and D.A. Kenny [50] was applied. In the first step, it was checked if the relationship between the independent variable and the dependent variable was statistically significant (path c). The second step involved determining if the relationship between the independent variable and the mediator (path a) and between the mediator and the dependent variable (path b) was statistically significant. In the third step, it was checked whether the relationship between the independent variable and the dependent variable weakened following the introduction of the mediator into the regression model. A relation diagram in the mediation model is shown in Figure 2.

The next stage of the analysis was to identify the relationship between positive orientation and tiredness/fatigue involving the emotional control mediator in the general dimension. The first step confirmed a significant relationship between positive orientation and tiredness/fatigue occurring in the students under study (excluding the mediator) (path c). The regression model proved to be well-fitted to the data (F(1.892) = 113.88, *p* < 0.0001, β = −0.336, *p* < 0.0001), and indicated an averagely strong significant relationship between the variables. Therefore, the obtained result confirmed that positive orientation was a predictor of tiredness/fatigue, which means that the nursing students with a higher level of positive orientation experienced lower levels of tiredness/fatigue. The second step of the analysis tested the relationship between the independent variable, i.e., positive orientation and the emotional control in the general dimension (the mediator) (path a). The regression model proved to be well-fitted to the data (F(1.892) = 99.583, *p* < 00000, β = −0.317, *p* < 0.0001), and indicated an averagely strong significant negative relationship. The analysis results confirmed that positive orientation was also a predictor of emotional control. The higher the positive orientation result, the weaker the suppression of negative emotions. The relationship between the mediator and the dependent variable, i.e., tiredness/fatigue (path b), was then tested. The regression model also proved to be well-fitted to the data and indicated a statistically significant positive relationship with low strength (F(1.892) = 37.603, *p* < 0.00001, β = 0.201, *p* < 0.0001), which means that the greater the tendency toward inhibiting (suppressing) negative emotions, the higher the degree of tiredness/fatigue experience by students. In the third step, a mediator was introduced into the model, and the level of the mediatory model’s significance was assessed. The strength of the relationship between positive orientation and tiredness/fatigue following the introduction of the mediator decreased slightly but remained statistically significant (β = −0.303, *p* < 0.0001) (path c’). This indicates the occurrence of partial mediation. It can be said that the mediator “captures” a portion of the independent variable’s effect on the dependent variable. The conducted analysis demonstrated significant partial mediation confirmed by the Sobel test result (z = −5.35, *p* < 0.0001). A graphical representation of the obtained results is shown in Figure 3.

Subsequent statistical analyses were directed at investigating the mediating role of anger control in the relationship occurring between positive orientation and tiredness/fatigue. The first mediation condition was satisfied, as the direct relationship between positive orientation and tiredness/fatigue (path c) was confirmed earlier. The regression model proved to be well-fitted to the data (F(1.892) = 113.88, *p* < 0.0001, β = −0.336, *p* < 0.0001), and indicated an averagely strong significant negative relationship. The second step of the analysis checked the relationship between positive orientation and anger control as a mediator. The regression model also proved to be well-fitted to the data (F(1.892) = 21.741, *p* < 0.0001, β = −0.154, *p* < 0.0001), and indicated a significant negative direction with low strength (path a), which implies that students with a higher level of positive orientation exhibit a lower tendency to inhibit (suppress) anger. It was then attempted to clarify whether the mediator was statistically significantly related to the dependent variable. It appeared that the strength of the effect of anger control on tiredness/fatigue was very weak but statistically significant (β = 0.087, *p* < 0.0001) (path b). In the third step, the mediator was introduced into the model, and the level of the mediatory model’s significance was assessed. The strength of the relationship between positive orientation and tiredness/fatigue following the introduction of the mediator (anger control) decreased to its level β = −0.330 and proved to be statistically significant (*p* < 0.0001) (path c’) (Figure 4). Therefore, partial mediation of the anger control of the relationship between positive orientation and tiredness/fatigue was demonstrated, which was also confirmed by the Sobel test result (z = −2.29, *p* < 0.01).

The mediatory function of revealing depression in the relationship occurring between positive orientation and tiredness/fatigue was also analyzed. The first step confirmed a direct relationship between positive orientation and tiredness/fatigue (path c). The regression model proved to be well-fitted to the data (F(1.892) = 113.88, *p* < 0.0001, β = −0.336, *p* < 0.0001), and indicated an averagely strong significant negative relationship. The second step of the analysis tested the relationship between the independent variable, i.e., positive orientation and depression control (the mediator) (path a). The regression model proved to be well-fitted to the data (F(1.892) = 147.64, *p* < 0.0001, β = −0.376, *p* < 0.0001) and indicated an averagely strong significant negative relationship indicating the students’ subjective belief in their ability to control responses when experiencing negative emotions. The relationship between the mediator (depression control) and the dependent variable (tiredness/fatigue) (path b) was tested as well. The regression model proved to be well-fitted to the data (F(1.892) = 90.597, *p* < 0.0001, β = 0.303, *p* < 0.0001) and a statistically significant average relationship was obtained (path b). This means that with an increase in the level of subjective depression control, the tiredness/fatigue experienced by nursing students increases. In the third step, a mediator was introduced into the model, and the level of the mediatory model’s significance was assessed. In the model including both the independent variable and the mediator (path c’), the role of positive orientation decreased (β = −0.258, *p* < 0.0001) while maintaining the statistical significance level. As a result of the analysis, the Sobel test value proved to be statistically significant (z = −7.6, *p* < 0.0001), thus confirming the partially mediatory role of depression control in the “effect” of positive orientation on the degree of tiredness/fatigue. A list of the obtained β indices is provided in Figure 5.

The final mediation analysis was aimed to determine if anxiety control plays a significant mediating role between positive orientation and tiredness/fatigue revealed among nursing students. An averagely strong, negative, statistically significant relationship between positive orientation and tiredness/fatigue (path c) was confirmed. The regression model proved to be well-fitted to the data (F(1.892) = 113.88, *p* < 0.0001, β = −0.336, *p* < 0.0001). In the second step, the relationship was tested between positive orientation and anxiety control, i.e., the mediator (path a), which was also statistically significant, with low strength and a negative direction (F(1.892) = 52.696, *p* < 0.0001, β = −0.236, *p* < 0.0001). In a subsequent regression analysis, the mediator (anxiety control) indicated an averagely strong, significant, positive correlation with tiredness/fatigue with tiredness/fatigue (F(1.892) = 8.211, *p* < 0.004, β = 0.095, *p* < 0.0004) (path b). In the third step, an attempt was made to introduce the mediator into the model. However, the tendency to inhibit (suppress) anxiety did not prove to be a significant mediator of the relationship between positive orientation and tiredness/fatigue occurring in students, and this implies a lack of a mediating effect.

## 4. Discussion

Sudden public health emergencies cause serious social damage, affecting people all over the world. Students are a social group which is sometimes ill-prepared to cope adaptively in crisis situations [52,53]. Considerable changes are observed in various stages of human development [54,55]. In the case of young people, such periods include the time when they start studying at universities and, in consequence, they take on new challenges, sometimes associated with stress, anxiety and adaptive issues. The COVID-19 pandemic is a new situation, which is also difficult in academic circles. Therefore, it has an impact on students’ responses, emotional control and social behavior. Unexpressed negative emotions, repeated and persisting for a long time, can become the basis for fatigue, whereas positive orientation as a personal resource can play a significant role in maintaining good health [17].

Positive orientation has been found to correlate negatively with fatigue in students. The study findings suggest that positive orientation has much in common with one’s health status, especially in its emotional dimension.

Alessandri et al. argue that positive orientation is a multidimensional construct of a higher order, integrating three components, which—according to literature reports—are associated with well-being and success in many areas of human life, such as health, learning achievements and work performance [56]. The mean positive orientation score, as measured in this study, was 26.87 (SD = 5.90) points, which was much lower than in the normalization sample (M = 29.19; SD = 4.55) [16], as well as in an international study among students in Spain (M = 29.20; SD = 5.18) and Slovakia (M = 29.22; SD = 5.49) [57]. The findings of this study of positive orientation showed its level to be low in more than half of the students, which may result in unfavorable self-assessment, low life satisfaction and poor assessment of the chances of achieving personal goals, as well as negative perception of the capacity to cope with difficulties and adversities. This implies that further studies of this construct should be conducted, as positive orientation can be described as a personal resource, and, maybe, even as a meta-resource, i.e., a supra-resource, which optimizes the function of others [58]. Many researchers present positive orientation in terms of potential (which enables optimal functioning), predispositions (which largely contribute to adaptation and achievements in various areas of life) and personal traits (which is an element of human capital or individual adaptive resources) [5,14,59]. Positive orientation, regarded as one of the resources, can probably contribute to inducing commitment, and, by buffering the negative impact of anxiety, anger and depression, reduce the probability of the occurrence of fatigue [60].

The current study demonstrated positive orientation in the group of Polish nursing students to be correlated negatively with emotional control. According to the authors, this is particularly important in light of scientific reports showing correlations between emotional expression and various diseases. Therefore, it is desirable to reinforce positive orientation because its stronger sense (including self-esteem, optimism and life satisfaction) is associated with stronger expression of negative emotions and its weaker sense is associated with their suppression. The results of studies of positive orientation show that these postulates are reasonable.

The findings of a study by Tho et al. showed that, the more positive the students’ attitude, the more satisfied they were with college life quality [61]. Students are also more successful in their studies and their social life, because positivity helps them to perceive themselves as individuals capable of coping with challenges associated with the demands of academic life [62]. A study conducted by Brisesette et al. showed that students who demonstrated a higher optimism level at the start of a semester suffered from a smaller increase in the level of stress and depression compared with the rest of the group [63]. This may show that optimism also supports better preparation for stressful events. A study by Jurisevic et al. showed that students’ general positive orientation towards life and the future was correlated positively, but negatively with catastrophism, which implies that general positivity is correlated with frequently applying more adaptive strategies [64]. Similarly, Carver et al. found applying emotional regulation strategies in general stressful situations correlated positively with optimism and perceived control of a stressor [65]. Many studies have pointed out that the pandemic changed the conditions of life and studying [66,67], which resulted in experiencing negative emotions [68,69].

Nursing students surveyed in this study had a higher score for general emotional control than adults aged 20–30 years in normalization studies, whereas it was comparable to the mean values for two clinical groups—patients on dialysis and menopausal women [48]. The scores in the subscales for anger, depression and anxiety were compared with the respective mean scores in the normalization studies. Those for anger and depression, as measured in this study, were higher, and only that for the subjective anxiety control was lower. This means that students tended to suppress their negative emotions to an extent comparable to some clinical groups [48]. Further studies should provide more conclusive data.

The research conducted by Tavsanli et al. showed that the average scores for anger and anxiety control in a group of students were a little above the average level, which may suggest that students often suppress their emotions consciously and avoid showing them [70]. The literature review shows that to understand other people’s emotions, one should first understand one’s feelings and thoughts and then control and manage them [71]. Tavsanli et al. pointed to a significant diversity of the mean results in the general scale and in CECS subscales with respect to the study majors. Students of social sciences turned out to have achieved significantly higher scores than those of nursing, midwifery and physiotherapy and rehabilitation. These findings were similar to those observed in the studies by Bakioğlu and Begum et al. [72,73]. This may suggest that anxiety is a strong emotion which affects one’s behavior, attitudes and thoughts [74]. The research conducted by Malinowska-Lipień et al. showed that the majority of nurses used the emotional suppression mechanism as a way of coping with COVID-19 pandemic-related emotions [75]. This applied to the suppression of anxiety, depression and anger. The research conducted in Polish hospitals by Buchelt et al. showed that the pandemic had had a negative impact on the mental health of 38% of the nursing staff under study [76]. The epidemic situation and working with patients infected with SARS-CoV-2 was the major factor which intensified negative emotions. The most frequently experienced emotions included anger, helplessness, and anxiety about own and relatives’ lives and health [76,77,78,79,80,81]. When negative emotions are triggered by a negative situation, the ability to control them is the basic mechanism of coping with stress. The ability to respond properly when confronted with a threatening situation is a test of the ability to control emotions. A study conducted by Mocan et al. among nurses during the COVID-19 pandemic showed that individuals who experience anger at a higher level than usual when facing threatening situations often choose emotion-based strategies [77]. Similar findings were observed in a study by Bidzan et al. [18], conducted in Poland in March 2020, immediately after the state of pandemic was announced. Its findings showed that nurses adopted suppression mechanisms as a way of coping with negative emotions caused by work in pandemic conditions. These are alarming findings as suppressing emotional expression, especially during the common pandemic threat, intensifies them or brings about long-term effects, such as emotional tension with harmful consequences to one’s health [82]. Unexpressed emotions can contribute to developing neurotic disorders and psychosomatic diseases. Meanwhile, it seems obvious that nurses and nursing students having clinical activities at healthcare facilities work under stress caused by the pandemic threat, and, therefore, they should have unlimited access to psychological counselling. Studies conducted by other authors have shown that mental disorders were more intense among employees with limited access to education and psychological support. Properly organized training and classes in hospitals can help to reduce stress, and psychologists working in hospitals can provide appropriate support [77,78,83]. Diversification of educational methods at schools by taking into account the negative outcomes of the pandemic affect the students’ emotions and their adaptation to professional life. The students’ anxiety is exacerbated by their uncertainty about when they will return to their “normal” lives. Although institutions make changes in education and training in a variety of ways, the top priority should be given to gaining students’ trust through active communication [84]. With the long-term nature of the COVID-19 pandemic, many people have experienced some form of fatigue, either physical or mental [85]. Students also suffered from sudden and intense life stressors [79,80], being exposed to a higher risk of depressive symptoms. Fatigue is often an effect of the exhaustion of mental resources, and its level can be affected by risk factors associated with stress and suppressing emotions [81,86]. An important role in fatigue severity is played by protective factors, described as personal resources, because they can adequately alleviate symptoms of physical and mental fatigue [87].

This study showed the fatigue score to be 37.93 (SD = 11.38) points, indicating clinically significant fatigue among Polish nursing students. In the study by Sajadi et al., only 13.1% of students claimed that they had not experienced fatigue, whereas the other respondents suffered from moderate and strong fatigue [88]. Research conducted by Nojomi et al. [89], Lowry et al. [90], Amaducci et al. [35] and Lai et al. [91] showed that over half of the students experienced moderate to severe fatigue, which is consistent with the findings of the above study [89]. A study conducted by Eslami [92] showed that half of the students had experienced fatigue. An interesting study on “mental fatigue” has been published recently by Morgul et al. [93]. The authors concluded that although the knowledge, attitudes and behavior concerning the prophylactic measures for COVID-19 are important for the prevention of disease transmission, they require commitment, which is associated with the participants’ fatigue. This, in turn, can lead to mental burden, e.g., anxiety associated with the pandemic [93,94].

Mosleh et al. reported that students attending online classes during the COVID-19 pandemic-related lockdown claimed to experience fatigue of moderate severity [95]. A considerable proportion of the students claimed to experience fatigue always, more often than five days a week. Possible reasons for such findings are consistent with earlier studies [96,97,98]. They constitute an accumulated mental and physical burden in personal, social and academic spheres. The results show that the students perceived the e-learning process as burdensome, which may have a negative impact on their mental state, fatigue and learning achievements. Among the various fatigue symptoms, participants in the study by Labraque et al. reported physical tiredness or exhaustion, headache and body pain, decreased motivation and increased worry as the most distinctive symptoms [30]. Similar symptoms of fatigue caused by lockdowns were mentioned in the report prepared by the Australian Psychological Association, which included sadness, physical exhaustion, decreased interest in previous activities and emotional outbreaks, as well as anxiety and fear [34].

This study has confirmed the direct correlation between general emotional control and anger, depression and anxiety control and fatigue. The study authors conclude that, with time, people become increasingly tired of the effects of the COVID-19 pandemic, and a chronically increased level of fatigue can have an adverse impact on the physical, mental, behavioral and cognitive functions of an individual. This was confirmed by other researchers, so it is extremely important to develop a strategy for solving the issue using scientific, evidence-based methods [99,100,101].

### Limitations and Implications Regarding Professional Practice

The considerations in this paper have implications for professional practice. First, the study results show that the fatigue experienced by nursing students in Poland during the COVID-19 pandemic requires effective actions aimed to provide support for their lives and health. Secondly, it is important to reinforce the role of positive orientation in students because the higher it is, the stronger the negative emotion expression, and the lower it is—the more suppressed it is. Thirdly, it is important for students to benefit from social and psychological support. At the academic level, there is a need for programs of mental health promotion and fatigue prophylaxis, including lifestyle-related factors.

This study has some limitations, because of which one should be careful in interpreting the findings. The main limitation is associated with the lack of data on the level of positive orientation, fatigue and the level of emotional control, including anger, anxiety and depression among nursing students in Poland during the period before the pandemic, which is important for comparison of the current level of the variables under study. Despite the limitations, this study provides important data, and offers a starting point for extended research on emotional control and fatigue and the role of positive orientation among nursing students.

## 5. Conclusions

A positive orientation as a personal resource was shown to be a predictor of tiredness/fatigue occurring in students during the COVID-19 pandemic. Students with a higher level of positive orientation experienced tiredness/fatigue to a lesser extent.

Emotional control as a general dimension, and in the categories of anger and depression control, was shown to be a partial mediator that reduced the strength of the relationship between positive orientation and tiredness/fatigue occurring in the students under study.

The suppression of anxiety in the analyzed mediatory model did not reduce the strength of the relationship between positive orientation and tiredness/fatigue occurring in the students under study and did not appear to be a mediator.

It is important to pursue prophylactic and preventive actions concerning tiredness/fatigue among nursing students and to reinforce positive orientation to enhance ability to control emotions (anger, depression and anxiety) to effectively minimize the effects of the COVID-19 pandemic.

## Figures and Tables

**Figure 1 jcm-11-02971-f001:**
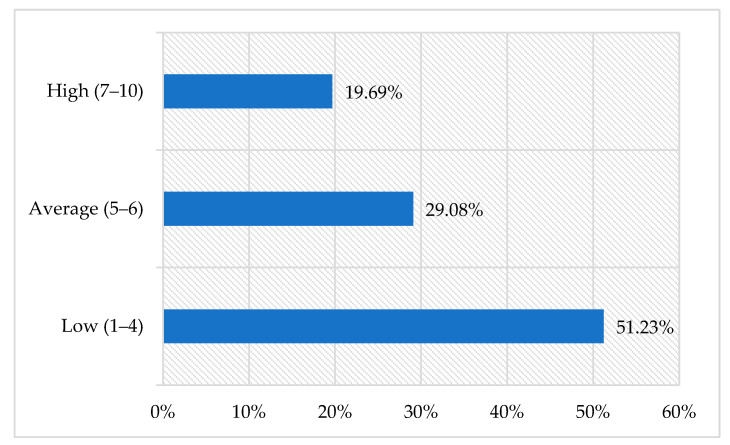
Positive orientation score structure among the nursing students on the sten scale.

**Figure 2 jcm-11-02971-f002:**
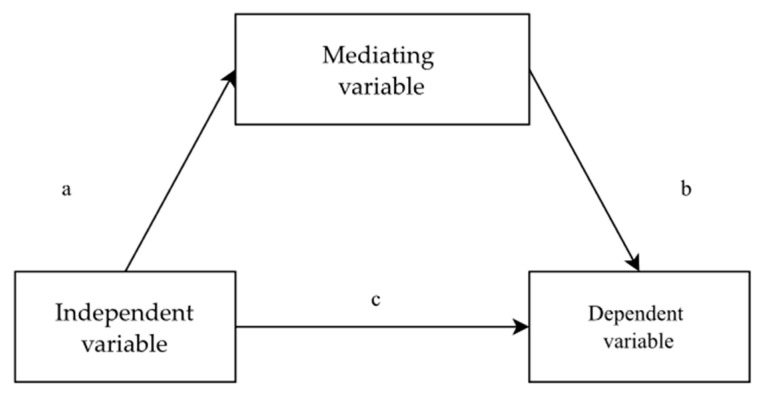
The relation diagram for the mediation model.

**Figure 3 jcm-11-02971-f003:**
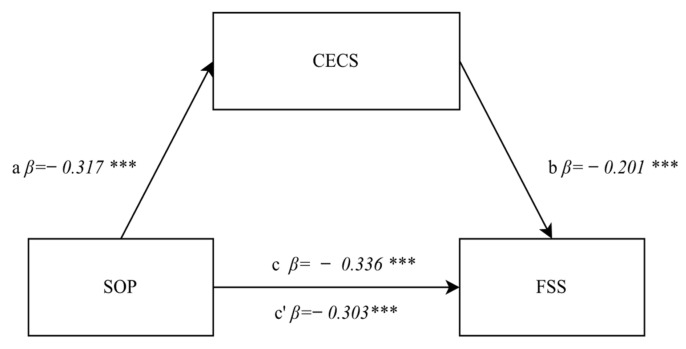
A relationship diagram in the model testing the mediatory effect of emotional control in the general dimension of the relationship between positive orientation and the degree of tiredness/fatigue. Statistically significant: *** *p* < 0.001. Explanation: SOP—positive orientation, FSS—fatigue severity scale, CECS—emotional control scale.

**Figure 4 jcm-11-02971-f004:**
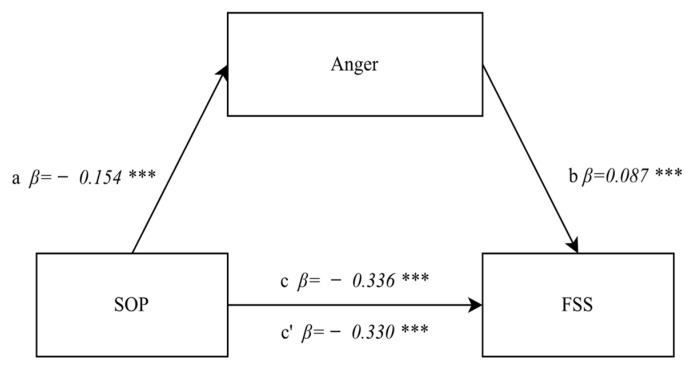
A relationship diagram of the model testing the mediatory effect of anger control on the relationship between positive orientation and the degree of tiredness/fatigue. Statistically significant: *** *p* < 0.001. Explanation: SOP—positive orientation, FSS—fatigue severity scale.

**Figure 5 jcm-11-02971-f005:**
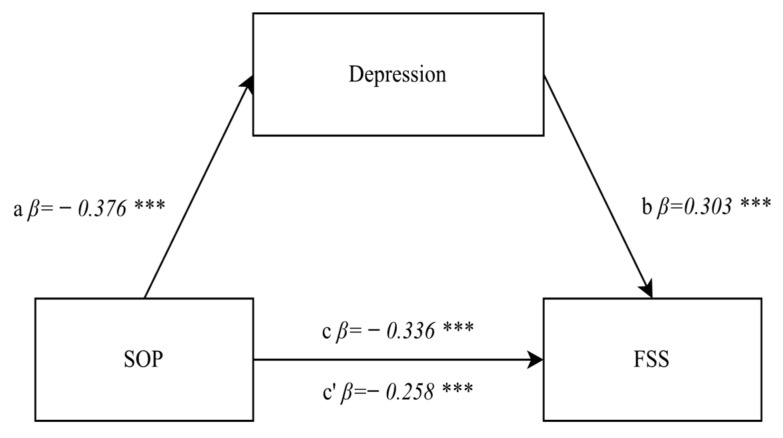
A relationship diagram of the model testing the mediatory effect of anxiety control of the relationship between positive orientation and the degree of tiredness/fatigue. Statistically significant: *** *p* < 0.001. Explanation: SOP—positive orientation, FSS—fatigue severity scale.

**Table 1 jcm-11-02971-t001:** Descriptive statistics of the variables under analysis.

Variables		N = 894						
		M	95% CI	Me	Min.–Max.	SD	Skewness	Kurtosis
SOP		26.87	26.48–27.26	27	9–40	5.90	−0.34	0.02
FSS		37.93	37.20–38.65	38	9–63	11.38	−0.04	−0.22
Subscales-CECS	Anger control	16.66	16.13–16.82	16	7–28	4.79	0.18	−0.74
	Depression control	17.87	17.91–19.58	19	7–29	4.62	−0.17	−0.54
	Anxiety control	16.92	16.71–17.36	17	7–30	4.85	0.03	−0.32
CECS-general emotional control		51.45	50.95–52.55	52	21–84	11.74	0.05	−0.16

Explanation: N—sample size, M—arithmetic mean, 95% CI—confidence interval of the mean, Me—median, Min.—minimum, Max.—maximum, SD—standard deviation, SOP—positive orientation, FSS—fatigue severity scale, CECS—emotional control scale.

**Table 2 jcm-11-02971-t002:** A matrix of Pearson correlation (r) coefficients between the variables under analysis.

	Variables		1.	2.	3.	4.	5.	6.
1.	SOP		-					
2.	FSS		−0.336 ***	-				
3.	Subscales-CECS	Anger control	−0.154 **	0.088 ***	-			
4.		Depression control	−0.376 ***	0.303 ***	0.508 ***	-		
5.		Anxiety control	−0.236 **	0.095 ***	0.408 ***	0.481 ***	-	
6.	CECS—General emotional control		−0.317 ***	0.201 ***	0.804 ***	0.824 ***	0.779 ***	-

Statistically significant: ** *p* < 0.01; *** *p* < 0.001; The analyses are presented for the number of observations N = 894. Explanation: SOP—positive orientation, FSS—fatigue severity scale, CECS—emotional control scale.

## Data Availability

The data presented in this study are available on request from the first author.

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
