# Peer review of "Positive Orientation and Fatigue Experienced by Polish Nursing Students during the COVID-19 Pandemic: The Mediatory Role of Emotional Control"

_jcm, 2022, doi:10.3390/jcm11112971_

Round 1

Reviewer 1 Report

I have reviewed the paper title: Emotional control and fatigue experienced by Polish nursing students during the COVID-19 pandemic: the mediatory role of positive orientation. The paper is reasonably written that are in line the scope of this journal issue. The relevant and contributes to existing literature. Therefore, I recommend this paper subject to the following revisions.

- Reorganise the contents of this manuscript to improve the flow and coherence for easy understanding. Where appropriate, insert sub-headings to segregate different points for clear illustration and discussion.
- In the abstract and introduction, clearly specify the research problems and research objectives. Please more clear about it. What's paper to clear research gap? please calrify.
- Introduction; You should indicate a gap in the previous research, or extend previous knowledge in some way.
- How to make sure you get the correct sample size?
- Suggest including a research framework covering all the relevant variables for clear illustration and easy understanding.
- Include a section on practical implications for possible adoption by the targeted audience. Should there be policy recommendations for policy maker / stakeholders?
- Ensure in-text citations are done in the correct format. Please refer to manuscript formatting guidelines.

This article is strongly recommended for publication after incorporating certain changes. This article needs thorough proofreading. Overall quality of Language is good. Just minor grammatical mistakes are found. All tables and figures are relevant. Research Methodology has been well defined. All data are aligned to the findings of the research. This article is good attempt in the field research and will be beneficial for future researchers.

Author Response

Dear Reviewers, Thank you very much for a thorough editorial assessment of my manuscript, positive opinions, as well as the reviewers’ remarks. I used them as an important guide to improving the quality of my paper. The implemented corrections were done strictly according to their comments. All changes made in the text are marked in red. I have enclosed the re-edited manuscript and cover letter as responses to Reviewers, detailing how I followed their suggestions. Thank you very much for your kind consideration of my paper. Yours sincerely, Ewa Kupcewicz, PhD 1. I have reviewed the paper title: Emotional control and fatigue experienced by Polish nursing students during the COVID-19 pandemic: the mediatory role of positive orientation. The paper is reasonably written that are in line the scope of this journal issue. The relevant and contributes to existing literature. Therefore, I recommend this paper subject to the following revisions. - We thank you for reviewing our paper. 2. Reorganise the contents of this manuscript to improve the flow and coherence for easy understanding. Where appropriate, insert sub-headings to segregate different points for clear illustration and discussion. - As recommended by the reviewer, the authors of the study have made extensive changes to the manuscript. The corrections have been highlighted in the manuscript. 3. In the abstract and introduction, clearly specify the research problems and research objectives. Please more clear about it. What's paper to clear research gap? please calrify. - In the section “Introduction”, the authors of the study attempted to clarify the background of the investigated problem and the justification of the study undertaken among nursing students. The aim of the study has been modified, and research questions have been introduced. The abstract has also been modified. The corrections have been highlighted in the manuscript. 4. Introduction; You should indicate a gap in the previous research, or extend previous knowledge in some way. - In the “Introduction” section, the authors explain why the undertaken study is important. The introduced modifications have been highlighted in the manuscript. 5. How to make sure you get the correct sample size? - At the study planning stage, a decision was made about the sample size (min. 560) while taking into account the statistical requirements so that our results, with a specified confidence level and an assumed maximum error, would estimate true results in the population. In the academic year 2020/2021, a total of 8,202 people studied nursing at the bachelor level in Poland (the size of the studied sample is 894, which accounts for over 10% of all the students). As the literature suggests, for single-element mediations (a single mediator), a sufficient sample should comprise approx. 75 units. MacKinnon D.P., Coxe S., Baraldi A.N., Guidelines for the investigation of mediating variables in business research, "Journal of Business Psychology" 2012, Vol. 27, pp. 1-14 6. Suggest including a research framework covering all the relevant variables for clear illustration and easy understanding. - The theoretical assumptions underlying the study being constructed have been verified, and the variables have been defined. 7. Include a section on practical implications for possible adoption by the targeted audience. Should there be policy recommendations for policy maker / stakeholders? - The subsection “Limitations and implications regarding professional practice”. 8. Ensure in-text citations are done in the correct format. Please refer to manuscript formatting guidelines. - The manuscript has been adapted to the JCM’s guidelines. 9. This article is strongly recommended for publication after incorporating certain changes. This article needs thorough proofreading. Overall quality of Language is good. Just minor grammatical mistakes are found. All tables and figures are relevant. Research Methodology has been well defined. All data are aligned to the findings of the research. This article is good attempt in the field research and will be beneficial for future researchers. - As suggested by the Reviewer, final linguistic corrections were carried out by the translation office, OSCAR Translations, ul. Reja 2/4 lok 1, 10-565 Olsztyn, Poland

Reviewer 2 Report

Thank you very much for inviting me to review this manuscript. The study analyzed the association between fatigue severity (FSS), emotional controls (CESC, anger, depression, and anxiety), and positive orientation (SOP) in 894 Polish nursing students during the COVID-19 pandemic. Specifically, this study treated positive orientation as a mediator.

The major concern is the causal inference description. Based on the authors’ assumption that positive orientation (SOP) was a mediator, the SOP should occur AFTER emotional controls (CESC, anger, depression, and anxiety). However, the level of positive orientation (SOP) could be a predictor of the expression and suppression of emotions (emotional controls). Thus, positive orientation (SOP) should be treated as a confounder rather than a mediator. The rationale for defining these variables as mediators or confounders is lacking, making the analysis invalid. Given that positive orientation (SOP) is a confounder, the manuscript may need a thorough revision.

Has the normality of the distribution of fatigue severity (FSS) in the studied samples been assessed?

It is unclear how the study subjects were recruited. Additionally, the representative of the study subjects should be clarified.

The authors used single sentences as a paragraph. Weak connection and flow make the manuscript difficult to follow. This manuscript should be English copyedited.

Author Response

Dear Reviewers, Thank you very much for a thorough editorial assessment of my manuscript, positive opinions, as well as the reviewers’ remarks. I used them as an important guide to improving the quality of my paper. The implemented corrections were done strictly according to their comments. All changes made in the text are marked in red. I have enclosed the re-edited manuscript and cover letter as responses to Reviewers, detailing how I followed their suggestions. Thank you very much for your kind consideration of my paper. Yours sincerely, Ewa Kupcewicz, PhD 1. Thank you very much for inviting me to review this manuscript. The study analyzed the association between fatigue severity (FSS), emotional controls (CESC, anger, depression, and anxiety), and positive orientation (SOP) in 894 Polish nursing students during the COVID-19 pandemic. Specifically, this study treated positive orientation as a mediator. - We thank you for reviewing our paper. 2. The major concern is the causal inference description. Based on the authors’ assumption that positive orientation (SOP) was a mediator, the SOP should occur AFTER emotional controls (CESC, anger, depression, and anxiety). However, the level of positive orientation (SOP) could be a predictor of the expression and suppression of emotions (emotional controls). Thus, positive orientation (SOP) should be treated as a confounder rather than a mediator. The rationale for defining these variables as mediators or confounders is lacking, making the analysis invalid. Given that positive orientation (SOP) is a confounder, the manuscript may need a thorough revision. - Following the reviewer’s guidance, profound changes have been introduced to the manuscript, involving the modification of the study aim and the introduction of research questions. Every effort has been made to make the “Introduction” section more valuable. An in-depth literature analysis has also been carried out. Following the reviewer’s guidance, the method for interpreting the study variables (positive orientation, tiredness/fatigue, and emotional control) has been clarified. A consultation has been held with experienced statisticians (employees of the Faculty of Mathematics and Computer Science at the University of Warmia and Mazury in Olsztyn) who have been dealing with biostatistics for many years, and the selection of statistical tests and the correctness of the analyses conducted were both verified. Extensive changes have been made in the subsection “Mediation analysis”. As a result of the introduced changes, a modification of the study title has been proposed. 3. Has the normality of the distribution of fatigue severity (FSS) in the studied samples been assessed? - The normality of the fatigue severity scale (FSS) in the studied samples was assessed using the W Shapiro-Wilk test. The distribution is close to the normal distribution at a level W=0.99; p=0.001 4. It is unclear how the study subjects were recruited. Additionally, the representative of the study subjects should be clarified. - The study participant recruitment method has been clarified in the subsection “ 2.1. Settings and design”. 5. The authors used single sentences as a paragraph. Weak connection and flow make the manuscript difficult to follow. This manuscript should be English copyedited. -As suggested by the Reviewer, final linguistic corrections were carried out by the translation office, OSCAR Translations, ul. Reja 2/4 lok 1, 10-565 Olsztyn, Poland

Reviewer 3 Report

Dear authors,

Your work is an important contribution to understanding the impact of the COVID 19 pandemic on the psychological well-being of nursing students.

The abstract should be unstructured, thus numbers 1 - 4 should be removed.

The results should be written more succinctly, without repeating what is written in the subtitle, with no explaining the importance of the result obtained.

The discussion is too long with too many descriptions of the studies you cited. Because of that the interpretation of your results does not come forward in its full right.

The order of references does not correspond to their appearance in the text - in the results there is reference 61 and in the discussion reference 86 appears instead of reference 76.

The number of cited papers is too large and has to be reduced.

Author Response

Dear Reviewers, Thank you very much for a thorough editorial assessment of my manuscript, positive opinions, as well as the reviewers’ remarks. I used them as an important guide to improving the quality of my paper. The implemented corrections were done strictly according to their comments. All changes made in the text are marked in red. I have enclosed the re-edited manuscript and cover letter as responses to Reviewers, detailing how I followed their suggestions. Thank you very much for your kind consideration of my paper. Yours sincerely, Ewa Kupcewicz, PhD 1. Your work is an important contribution to understanding the impact of the COVID 19 pandemic on the psychological well-being of nursing students. - We thank you for reviewing our paper. 2. The abstract should be unstructured, thus numbers 1 - 4 should be removed. - Items 1-4 have been removed from the abstract. 3. The results should be written more succinctly, without repeating what is written in the subtitle, with no explaining the importance of the result obtained. - In the “Introduction” section, the authors of the study attempted to clarify the background of the investigated problem and the justification of the study undertaken among nursing students. The aim of the study has been modified, and research questions have been introduced. The abstract has also been modified. Following consultation with statisticians, extensive changes have been made in the “Results” section. A modification to the title of the study has been proposed. The corrections have been highlighted in the manuscript. 4. The discussion is too long with too many descriptions of the studies you cited. Because of that the interpretation of your results does not come forward in its full right. - The authors of the study have made every effort and introduced amendments to make the “Discussion” section more valuable. The corrections have been highlighted in the manuscript. 5. The order of references does not correspond to their appearance in the text - in the results there is reference 61 and in the discussion reference 86 appears instead of reference 76. -The order of the introduced references has been verified. The corrections have been highlighted in the manuscript. 6. The number of cited papers is too large and has to be reduced. - The number of the cited papers has been verified, and those less relevant have been removed. The corrections have been highlighted in the manuscript.

Round 2

Reviewer 2 Report

Thank you very much for the response and revision. I appreciate that the analytical framework has been substantially revised by treating emotional controls (CESC, anger, depression, and anxiety) as mediators. However, most of my concerns remain.

The major concern regarding the causal inference description is partially solved based on the new analytical framework. However, whether the status of positive orientation (SOP) existed BEFORE the observed level of emotional controls (CESC, anger, depression, and anxiety) should be clarified. That is, the dependent variable should occur BEFORE the mediating variable, followed by the dependent variable. The point in time at which the state of each variable should be explicitly examined.

Based on the result of the Shapiro-Wilk test, the p-value is 0.001, indicating the variable is not normally distributed. It is unclear how the distribution is close to the normal distribution. It should be justified. Additionally, the results should be reported in the manuscript.

The question regarding “It is unclear how the study subjects were recruited. Additionally, the representative of the study subjects should be clarified” was not adequately addressed. The text in “2.1. Settings and design” did not address “representativeness,” so I suggested clarifying this point in the revision.

There are still single sentences as a paragraph in the revision. A careful check is needed.

Author Response

Dear Reviewer, Thank you very much for your comments. They have been used as an important guide to improve the quality of the manuscript. All changes made to the text are marked in red. We would like to explain that the empirical data used in the work is part of a larger research project carried out by a team of researchers, which includes four professors who overlook the methodological correctness of the research. One of the professors is a clinical psychologist with scientific achievements in the studied discipline. The presented work has a great scientific potential and the obtained results can certainly be implemented in professional practice. We kindly ask you to take into account our explanation and admit our manuscript to further stages of the procedure. Yours sincerely, Ewa Kupcewicz PhD REV. II 1. Thank you very much for the response and revision. I appreciate that the analytical framework has been substantially revised by treating emotional controls (CESC, anger, depression, and anxiety) as mediators. However, most of my concerns remain. - We thank you for reviewing our paper. 2. The major concern regarding the causal inference description is partially solved based on the new analytical framework. However, whether the status of positive orientation (SOP) existed BEFORE the observed level of emotional controls (CESC, anger, depression, and anxiety) should be clarified. That is, the dependent variable should occur BEFORE the mediating variable, followed by the dependent variable. The point in time at which the state of each variable should be explicitly examined. - The authors of the study explain: Positive orientation is the basic tendency to notice and attach importance to the positive aspects of life, experiences and yourself. It is a higher-order latent variable that combines three components: self-esteem, optimism and life satisfaction. The first two components have the status of a personality trait, i.e. they are relatively persistent, rarely changeable over the course of life, and are formed early in life. They are largely temperamental, which indicates their biological conditioning. For this reason, there is no doubt that positive orientation is more constant and underlying, and that it existed before CECS - that is the theory. Caprara, G. V. Positive orientation: Turning potentials into optimal functioning. The Bulletin of the European Health Psychologist 2009, 11(3), 46–48. 3. Based on the result of the Shapiro-Wilk test, the p-value is 0.001, indicating the variable is not normally distributed. It is unclear how the distribution is close to the normal distribution. It should be justified. Additionally, the results should be reported in the manuscript. - The authors consulted an experienced statistician and verified the selection of statistical tests and the correctness of the analyzes performed. The similarity of the distribution of the discussed variables to the normal distribution was tested again, consisting of determining the values of skewness and kurtosis. On the basis of the literature, referring to the central limit theorem, it is assumed that the distribution of the results of individual variables will be a normal distribution, when the skewness and kurtosis of the examined variables will be in the range from -1.5 to 1.5. The results of the analysis show (tab.) that the variables have a normal distribution. Variables N; Skewness; Kurtosis Emotion control indicator (CESC), 894, 0.05, -0.16 Anger 894, 0.18, -0.74 Depression 894, -0.17, -0.54 Anxiety 894, 0.03, -0.32 FSS (Tiredness) 894, -0.04, -0.22 SOP (Positive orientation) 894, -0.34, 0.02 BedyÅ„ska S., CypryaÅ„ska M. Statystyczny drogowskaz 1. Praktyczne wprowadzenie do wnioskowania statystycznego. SzkoÅ‚a Wyższa Psychologii SpoÅ‚ecznej. Warszawa 2012. 4. The question regarding “It is unclear how the study subjects were recruited. Additionally, the representative of the study subjects should be clarified” was not adequately addressed. The text in “2.1. Settings and design” did not address “representativeness,” so I suggested clarifying this point in the revision. - At the study planning stage, a decision was made about the sample size (min. 560) while taking into account the statistical requirements so that our results, with a specified confidence level and an assumed maximum error, would estimate true results in the population. In the academic year 2020/2021, a total of 8,202 people studied nursing at the bachelor level in Poland (the size of the studied sample is 894, which accounts for over 10% of all the students). The study was conducted in several cities, taking into account the distribution of universities educating students in nursing in various regions of Poland. The sample created in this way can be treated as a representative sample for the population of students studying nursing in Poland. 5. There are still single sentences as a paragraph in the revision. A careful check is needed. - We agree. Corrections were introduced to the manuscript.

Reviewer 3 Report

Dear Authors,

after the correction according to the instructions of the reviewers I have no major objections on your article "Positive Orientation and Fatigue Experienced by Polish Nursing Students During the Covid-19 Pandemic: the Mediatory Role of Emotional Control ".

Author Response

Thank you very much